# Nurses’ Opinions on Frailty

**DOI:** 10.3390/healthcare10091632

**Published:** 2022-08-26

**Authors:** Robbert J. Gobbens, Sofie Vermeiren, An Van Hoof, Tjeerd van der Ploeg

**Affiliations:** 1Faculty of Health, Sports and Social Work, Inholland University of Applied Sciences, 1081 HV Amsterdam, The Netherlands; 2Zonnehuisgroep Amstelland, 1186 AA Amstelveen, The Netherlands; 3Department Family Medicine and Population Health, Faculty of Medicine and Health Sciences, University of Antwerp, 2610 Antwerp, Belgium; 4Department of Health and Science, AP University of Applied Sciences and Arts Antwerp, BE-2000 Antwerp, Belgium

**Keywords:** frailty, opinions of nurses, older people, holistic approach, prevention

## Abstract

Nurses come into frequent contact with frail older people in all healthcare settings. However, few studies have specifically asked nurses about their views on frailty. The main aim of this study was to explore the opinions of nurses working with older people on the concept of frailty, regardless of the care setting. In addition, the associations between the background characteristics of nurses and their opinions about frailty were examined. In 2021, members of professional association of nurses and nursing assistants in the Netherlands (V&VN) received a digital questionnaire asking their opinions on frailty, and 251 individuals completed the questionnaire (response rate of 32.1%). The questionnaire contained seven topics: keywords of frailty, frailty domains, causes of frailty, consequences of frailty, reversing frailty, the prevention of frailty, and addressing frailty. Regarding frailty, nurses especially thought of physical deterioration and dementia. However, other domains of human functioning, such as the social and psychological domains, were often mentioned, pointing to a holistic approach to frailty. It also appears that nurses can identify many causes and consequences of frailty. They see opportunities to reverse frailty and an important role for themselves in this process.

## 1. Introduction

Frailty is a concept that is receiving plenty of attention from science, politics, and healthcare practice. This is not surprising, since frailty is related to ageing [1], and the proportion of older people in the population will increase worldwide [2]. In addition, many studies have showed that frail older people have a high risk of adverse outcomes, such as disability in performing activities of daily living [3], increase in healthcare use (hospitalisation, institutionalisation) [4,5], lower quality of life [6], and premature death [7,8]. Thus, the importance of identifying frail older people is evident. This offers opportunities to reduce frailty and/or prevent or at least delay the adverse outcomes mentioned above.

However, there is no consensus on the definition of frailty. Currently, many conceptual and operational definitions of frailty exist with associated measurement tools. An important point of disagreement is whether frailty refers exclusively to the physical functioning of older people or whether other domains of human functioning should also be considered, such as the psychological and social domains. The most cited measurement tool, ‘the phenotype of frailty’, refers to the physical functioning of older people [9]. This tool contains five criteria that can be used to determine whether a person is non-frail, pre-frail, or frail: unintentional weight loss, weakness, slowness, low physical activity, and poor endurance [9]. A more recently developed instrument, the FRAIL scale [10] questionnaire, only includes physical limitations that older people may have. 

Another widely used instrument is the Frailty Index (FI), based on the Canadian Study of Health and Aging (CSHA) Cumulative Deficit Model. This includes other domains of human functioning, chronic diseases, and items referring to disability in performing activities of daily living [11]. In recent years, these so-called multidimensional instruments for determining frailty have been on the rise. One other example is the Tilburg Frailty Indicator (TFI), a self-report questionnaire, [12] which includes a physical, psychological, and social domain. Because the definitions and, therefore, the measurement instruments of frailty differ, it is not clear what the causes and consequences of frailty are. In this way, the FI considers disability in performing activities of daily living and chronic diseases as components of frailty [11], while the phenotype of frailty disregards these two concepts [9]. 

While it is recognised that beliefs of healthcare professionals have an impact on the identification, including screening, and intervention in frail older people, few studies have focused on their opinions of the concept of frailty. In a qualitative study among 33 Italian primary care professionals (family physicians, nurses, social workers, physiotherapists, home care workers), three main themes emerged: the psychosocial nature of frailty, late detection of frailty, and barriers to feasible frailty prevention [13]. Another qualitative study including 22 community care professionals underscored the need of a holistic interdisciplinary approach for the benefit of the care of frail older people, as well as a further exploration of the interacting nature of the components of frailty, particularly the role of modifiable factors [14]. Swedish healthcare professionals also point to a holistic approach to frailty, including components such as having an inadequate social network and being negatively influenced by personal qualities [15]. Among 31 healthcare professionals, using focus groups, it was concluded that the education and training of these professionals and interdisciplinary collaboration were essential for the delivery of person-centred care for frail community-dwelling older people [16]. 

Nurses come into frequent contact with frail older people in all healthcare settings (e.g., primary care, hospital, nursing home). However, few studies have specifically asked nurses about their views on frailty [17,18,19]. Uncertainty regarding the definition of frailty was present among six community nurses; this was particularly true for nurses who had received limited frailty training [17]. In Scotland, 18 community nurses described frailty as vulnerability, loss, and complex comorbidity. According to these nurses, their current practice was largely reactive and influenced by both professional judgment and intuition. In addition, there was little systematic screening and assessment of frailty in older people [18]. A Dutch study, using semi-structured interviews with nurses (*N* = 13) focusing on screening for frailty among older hospitalised people, found that most nurses believe that measurement instruments are helpful in daily practice. In particular, they appeared to be interested in specific components of frailty (e.g., fall risk) [19]. 

The aforementioned three studies focusing on nurses’ perceptions on frailty all had an exploratory qualitative design, using semi-structured interviews, focus groups characterised by small numbers of participants (max *N* = 18), and one care setting (community or hospital). The aim of this study was to explore the opinions of nurses working with older people on the concept of frailty, regardless of the care setting. In addition, the associations between the background characteristics of nurses and their opinions about frailty were examined. In contrast to previous studies, we used a quantitative design.

## 2. Methods

### 2.1. Study Population and Data Collection

V&VN is the professional association of nurses and nursing assistants in the Netherlands. About 100,000 nurses are members of this association. V&VN has several departments; the Geriatrics and Gerontology Department focuses specifically on the care of older people. This department has 781 members. In April 2021, all these members received a digital questionnaire asking their opinions on frailty.

### 2.2. Measurement

The questionnaire was originally developed in 2019 by scientists affiliated with AP University of Applied Sciences and Arts Antwerp, Belgium. They used focus groups, individual interviews, and the literature [20,21] to constitute the questionnaire [22]. To examine the face validity of the questionnaire and make it suitable for Dutch nurses, it was presented to five board members of the Geriatrics and Gerontology Department. Based on their comments, the questionnaire was slightly modified. The questionnaire contained the following seven main topics: keywords of frailty, frailty domains, causes of frailty, consequences of frailty, reversing frailty, prevention of frailty, and addressing frailty. The questionnaire also included eight questions on sociodemographic factors: gender, age, experience, hours, profession, institution, employed, and education. See Table 1 for the answering categories.

### 2.3. Statistical Analyses

For the description of the characteristics of the respondents, we used numbers and percentages. The response to the multiple-choice variables was described using ordered bar charts based on numbers. Association of the multiple-choice variables with the characteristics of the respondents was analysed using chi-square analysis. An asterisk (*) refers to answers mentioned more often than expected. A *p*-value <0.01 was considered significant. 

### 2.4. Ethics Approval

For this study, medical ethics approval was not necessary because particular treatments or interventions were not offered or withheld from respondents. The integrity of the respondents was not encroached upon as a consequence of participating in this study, which is the main criterion in medical–ethical procedures in the Netherlands [23].

## 3. Results

### 3.1. Characteristics of the Respondents

Of the 781 members of the V&VN Geriatrics and Gerontology Department, 251 individuals completed the questionnaire (response rate of 32.1%). Table 1 shows the characteristics of the respondents. In total, 94% were female. More than 50% of respondents worked ≥32 hours per week, and 37% were 55 years or older. Most respondents were employed in a hospital (41.8%). Approximately two-thirds of respondents were nurses (63.7%), 17.9% were nursing specialists, and 18.3% were ‘other’. Certified nursing assistants were included in the latter group. Henceforth, we refer to all respondents as nurses, unless otherwise noted.

### 3.2. Keywords of Frailty

Figure 1 shows the keywords the respondents most linked to frailty. According to the respondents, physical deterioration was most frequently linked to frailty, followed by dementia and dysfunction. Profession was associated with mentioning limitations in performing activities of daily living (*p*-value = 0.002, nursing specialist *), and institution was associated with mentioning loneliness (*p*-value = 0.002, primary healthcare *), chronic disease (*p*-value = 0.009, nursing home *), and limitations in performing activities of daily living (*p*-value = 0.001, hospital *). 

Figure 2 (the keywords of frailty part) presents the respondents’ answers to additional questions, which could be answered by yes or no. According to 13.5% of respondents, an individual could be considered frail if over 65 years of age. In addition, 59.4% of respondents believed that a person 80 years of age who was admitted to a hospital after a hip fracture was frail. The percentage was even higher (89.2%) if it involved someone aged 63 who lost his or her spouse, no longer had interests, and had unintentionally lost 7 kilograms.

### 3.3. Frailty Domains

Figure 3 shows the domains that the respondents associated with frailty. They believed that the physical domain belonged to frailty the most. There was no association with the characteristics of the respondents. All *p*-values were >0.01.

### 3.4. Causes of Frailty

Figure 4 presents the keywords the respondents most linked to possible causes of frailty. The three most frequently cited causes were, consecutively, memory problems, reduced mobility, and poor nutritional status. The age category was associated with reduced mobility (*p*-value = 0.005, age category <44 years *), and institution was associated with coping mechanisms (*p*-value = 0.000, nursing home *). 

### 3.5. Consequences of Frailty

Figure 5 shows the keywords most linked to possible consequences of frailty. The most frequently mentioned consequence was lower quality of life, followed by falling into a downward spiral and earlier death. Hours per week was associated with falls (*p*-value = 0.003, category <28 h *), and profession was associated with social isolation (*p*-value = 0.008, nurse *). 

### 3.6. Reversing Frailty

According to 76.9% of respondents, the process of frailty can be reversed (see the reversing frailty part of Figure 2). Figure 6 shows the factors that the respondents most linked to the possible influence on reversing the process of frailty. Respondents believed that optimising nutritional status has the most potential to reverse the process of frailty, followed by support from the environment and improving mobility. Institution was associated with support from the environment (*p*-value = 0, primary healthcare *). 

### 3.7. Prevention of Frailty

Figure 2 (the prevention of frailty part) presents the respondents’ opinions about the prevention of frailty. Nearly all respondents believed a preventive approach to frailty makes sense (98.0%). However, only 61% indicated that screening for frailty was conducted in their institution. Figure 7 presents the preventive measures mentioned by the respondents. The three most-cited preventive measures were network (involve family caregiver, family physician), exercise, and meaningful daytime activity. Education and exercise were significantly associated (*p*-value = 0.002, higher vocational education master *).

### 3.8. Addressing Frailty

Five yes/no questions were asked related to addressing frailty (see Figure 2). The vast majority of respondents (82.9%) indicated that frailty was treated in their institution. According to 76.9% of respondents, addressing frailty was part of their duties. They also thought this was the way it should be; 62.2% of respondents found it useful to address frailty by a specialised nurse, and 25.9% believed that another discipline was better suited to fulfil this task. 

## 4. Discussion

Due to the ageing population [2], frailty is an important issue in many countries around the world. Healthcare professionals in all care settings often have older people in care or treatment. This certainly applies to nurses. It is therefore of utmost importance to hear their opinions on frailty. This study aimed to explore the opinions of nurses regarding frailty. In addition, we examined the associations between background characteristics of nurses and their opinions about frailty. To achieve these aims, we used a questionnaire that focused on the following topics: keywords, domains, causes, consequences, reversing frailty, and prevention of frailty. The questionnaire was completed by 251 Dutch nurses. In this section, we discuss the most significant findings.

Regarding the keywords of frailty, physical deterioration was most frequently mentioned. This was not surprising, because frailty was originally a medical concept, and the emphasis was on the physical functioning of older people [24]. In addition, many measurement instruments mainly contain components that refer to physical limitations, such as the phenotype of frailty [9]. This also applies to the multidimensional instruments. For example, the TFI contains eight components belonging to physical frailty and only four and three components relating to psychological and social frailty, respectively [12]. Dementia was the participants’ second-most mentioned keyword. Like frailty, dementia is associated with disability and, furthermore, with a high burden of dependency [25]. A systematic review and meta-analysis revealed that individuals with frailty were at higher risk of dementia [26]. A more recent study concluded that frailty may play an inherent role in dementia, particularly in people aged 69 years and older [27]. Although frailty and dementia are related concepts, nurses must realise that they are not synonyms. 

The keyword limitations in performing activities of daily living refers to disability, and this was mentioned more by nursing specialists in comparison with nurses and others. Moreover, this keyword was mentioned more by nurses working in a hospital. Further analysis showed that it was mainly the nurse specialists who were associated with a hospital (51% of the nursing specialists worked in a hospital). This setting is usually aimed at the early discharge of patients. Being able to perform activities of daily living independently again is essential. In science, there is still debate as to whether limitations to performing activities of daily living belong to frailty or should be considered a possible consequence of it [28,29,30]. Loneliness was significantly more mentioned by nurses affiliated with primary healthcare (district nurses). This can be explained by the fact that they meet older people in their environment, and they often experience loneliness in older people. Although loneliness is associated with a lower quality of life of older people [31,32], this component is still often not included in the assessment of frailty [33]. In contrast, the keyword chronic diseases was more often mentioned by nurses working in nursing homes. The oldest elderly people live in nursing homes, and advanced age is associated with multimorbidity. As with limitations in performing activities of daily living, this is also the subject of scientific debate. Some scientists consider individual chronic diseases and multimorbidity as part of frailty [11,34], while others see these conditions as a determinant of frailty [35].

According to the participants, the physical domain of human functioning belonged most to frailty. This confirms what we found with the keywords. Not only do physical components always occur in instruments used to determine frailty, these components also have great predictive value for the occurrence of well-known adverse outcomes of frailty, such as disability, increased healthcare utilisation (e.g., hospitalisation, institutionalisation), and mortality, in comparison with psychological and social domains of frailty [36,37,38]. In addition to the physical domain, the cognitive domain was also frequently mentioned. Designating this domain separately refers to the discussion about cognitive frailty. The International Academy on Nutrition and Aging (IANA) and the International Association of Gerontology and Geriatrics (IAGG) defined cognitive frailty as the simultaneous occurrence of both mild cognitive impairment and physical frailty in which there is no dementia or other pre-existing brain disorders [39]. The key words also showed that nurses strongly associated frailty with cognitive limitations that older people may have. Moreover, many nurses believed that memory problems are the main cause of frailty. They also considered reduced mobility and poor nutritional status as main causes of frailty. However, in almost all measurement tools of frailty (e.g., phenotype of frailty, TFI, FI), these are components by which to determine whether someone is frail. Thus, nurses do not distinguish between components and causes of frailty. This was confirmed by the fact that financial problems, a known cause of frailty, were identified as a cause by few. To provide early intervention that can prevent or delay frailty, nurses must have knowledge of the causes of frailty.

Most nurses considered lower quality of life as a possible consequence of frailty. This is supported by many studies. A systematic review, including 11 cross-sectional studies and two longitudinal studies, and a meta-analysis of four cross-sectional studies, showed evidence of a consistent inverse association between frailty and quality of life in community-dwelling older people [6]. Interventions by nurses in frail older people should be aimed at keeping their quality of life as high as possible. Falling into a downward spiral and premature death were also two important possible consequences of frailty, according to nurses. As mentioned previously, much evidence is available for earlier death of frail older people [7,8]. 

According to most nurses, the process of frailty can be reversed. Studies have shown that this is possible [40]. Many nurses have the opinion that optimal nutritional status can influence or reverse the process of frailty. Since nutritional status is an important component of frailty, this result is not remarkable. Moreover, it is part of the nurse’s job to pay attention to nutrition, especially in frail older people. Improving mobility also scored high among respondents. A systematic review of primary care interventions, including 46 studies, showed that a combination of training for muscle strength and protein supplementation was the most effective intervention to reverse frailty [40]. Institution (primary healthcare) was associated with mentioning support from the environment as an influencing factor on reversing the process of frailty. District nurses care for frail elderly people at home and pay plenty of attention to the support that the environment can provide, such as informal care by neighbours or family members. An observational study demonstrated that exercise-based social participation was associated with reversing frailty, and older people who recovered from frailty had high individual-level social capital (e.g., interaction with neighbours, social participation) [41].

Because frail older people have a risk of disability, an increase in healthcare utilisation, and lower quality of life, it is important to focus interventions on the prevention of frailty. Almost all respondents believed that this made sense. To be able to conduct prevention properly, we need to screen for frailty. That did not appear to be the case in all healthcare institutions that the respondents were associated with. There are several validated and reliable instruments available in the Netherlands, such as the TFI [12] and the Groningen Frailty Indicator (GFI) [34]. On the topic of preventing frailty, nurses emphasised that the network was crucial. In their opinion, an important role was played by the informal network, such as involving family caregivers but also professionals, such as the general practitioner. However, a qualitative focus group study demonstrated that only a few general practitioners had actively carried out interventions to prevent frailty [42]. General practitioners as well as practice nurses experienced barriers to providing proactive and structured care to frail older people, such as lack of time and financial compensation [43]. Respondents who completed a master’s programme indicated significantly more, compared to the other groups, that physical exercise should be used primarily for the prevention of frailty. Physical exercise has been the subject of frailty research in many studies [44,45]. These highly trained nurses read more scientific studies and are therefore more knowledgeable about the contribution of exercise to the prevention of frailty.

Frailty was treated in most of the respondents’ institutions. This applies not only to hospitals but also to settings such as nursing homes and primary healthcare. In particular, the prevalence of frailty in nursing homes was high; a pooled estimate of the prevalence of frailty among nursing home residents was 52.3% (nine studies, *n* = 1373) [46]. However, it should be noted that frailty was predominantly measured physically, which likely provided a lower prevalence than if multidimensional measures had been used more frequently. Nurses perform an important role in treating frailty; with their qualities, they are able to meet the specific needs and desires that an individual frail elderly person may have. The starting point of nursing is a holistic human view, which considers the total functioning of human beings. Because frailty involves deficits in multiple domains of functioning, collaboration with other disciplines is essential (e.g., clinicians, physiotherapists, dieticians, occupational therapists, social workers). If there are deficits in the social functioning of older people, then collaboration with social workers plays a crucial role in managing frailty at the community level. Factors that influence this collaboration are the perception of the other professional, closeness of the other professionals, the perception of interdependence, and the frailty of the older person [47]. This requires interprofessional competencies that are being developed in many nursing education programmes. 

Some limitations of our study should be highlighted. Firstly, our questionnaire has not been extensively validated. Secondly, the questionnaire contained only closed questions; there was no room for the participants to add their own thoughts to the pre-structured answers. As a result, we decided to reduce the significance level of the *p*-value from 0.05 to 0.01. Finally, all participants were members of the Geriatrics and Gerontology Department of V&VN, which possibly led to selection bias.

Our study showed nurses’ opinions on frailty. It has become clear that nurses view frailty differently. What this study did not examine was the effect of this on the quality of care and outcomes in older people (e.g., an increase in healthcare utilisation, disability, lower quality of life, premature death). It would be interesting and important to conduct a study focused on the associations between nurses’ opinions about frailty and these outcomes. In addition, we recommend the development of an educational intervention based on the results of this study. This will potentially lead to a higher level of knowledge about frailty among nurses. Follow-up research can determine the effects of the developed educational intervention on adverse outcomes in older people, for example by performing a randomised controlled trial.

In this study we assessed the opinions of nurses regarding seven topics related to frailty: keywords of frailty, frailty domains, causes of frailty, consequences of frailty, reversing frailty, the prevention of frailty, and addressing frailty. When it comes to frailty, the nurses especially thought of physical deterioration and dementia. However, other domains of human functioning, such as the social and psychological domains, were often mentioned, pointing to a holistic approach to frailty, which was also highlighted in previous studies [14,15]. It also appears that the nurses could identify many causes and consequences of frailty. They saw opportunities to reverse frailty and an important role for themselves in this process. Some background characteristics of the nurses were associated with their opinions about the seven main topics of frailty. Institution (primary healthcare) was associated with two topics: mentioning loneliness as a keyword of frailty and reversing frailty with support from the environment.

## Figures and Tables

**Figure 1 healthcare-10-01632-f001:**
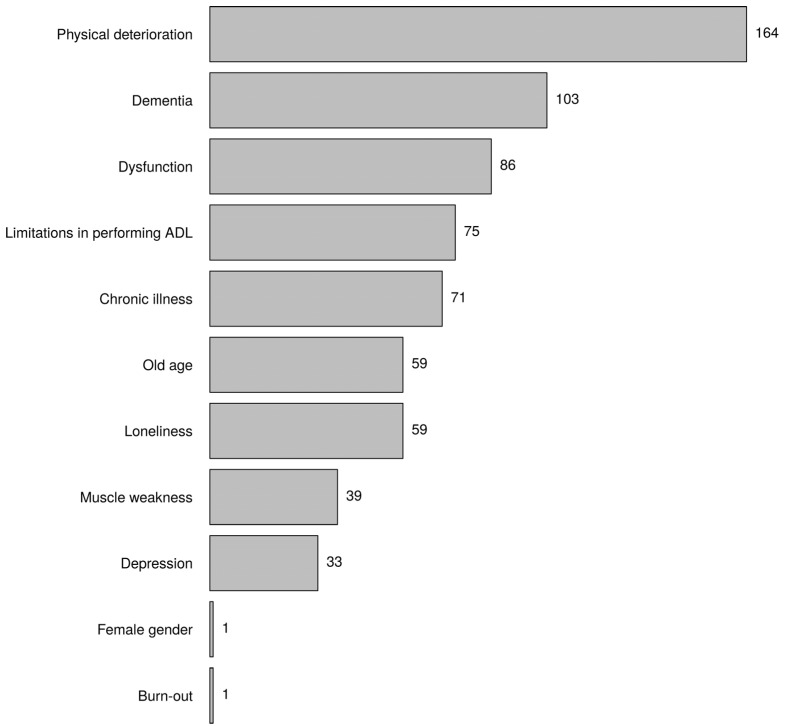
Indicate which three keywords you most link to the term ‘’frailty’. Choose three answers (the words that are most linked).

**Figure 2 healthcare-10-01632-f002:**
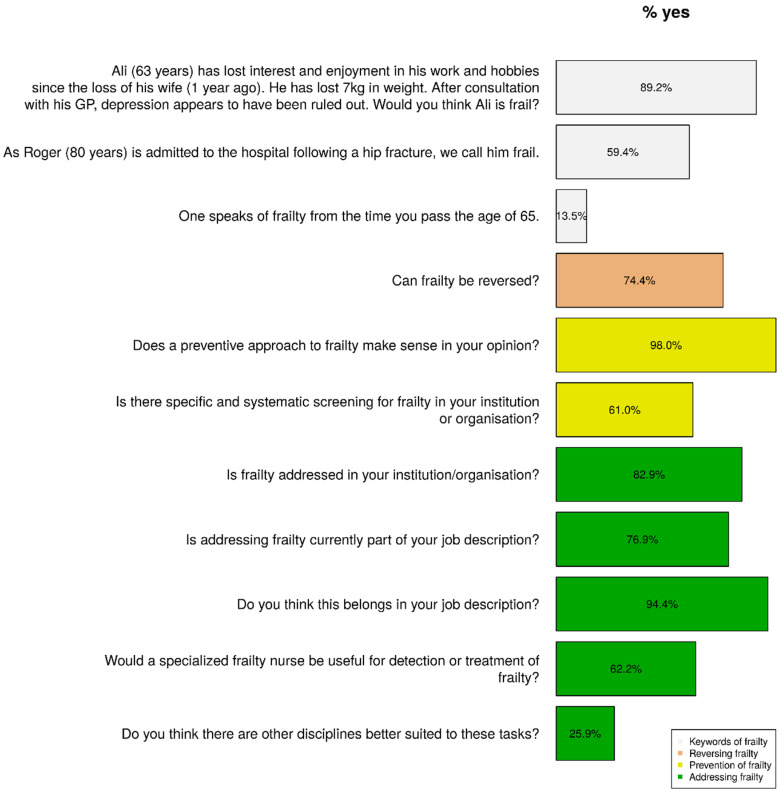
The ‘yes’ answers to keywords of frailty, reversing frailty, prevention of frailty, and addressing frailty.

**Figure 3 healthcare-10-01632-f003:**
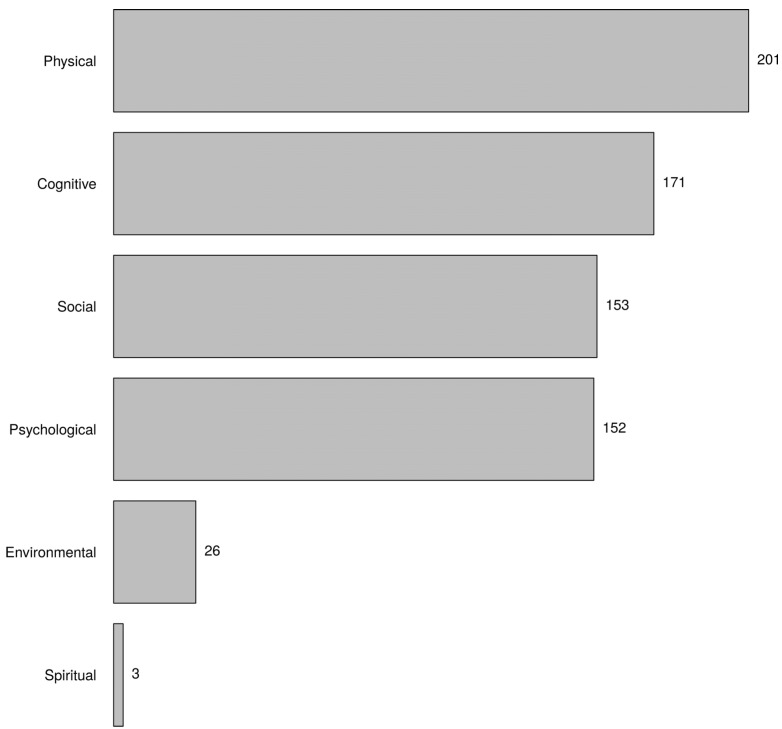
Which of the following domains do you think belong to frailty? Choose three answers (the most important domains).

**Figure 4 healthcare-10-01632-f004:**
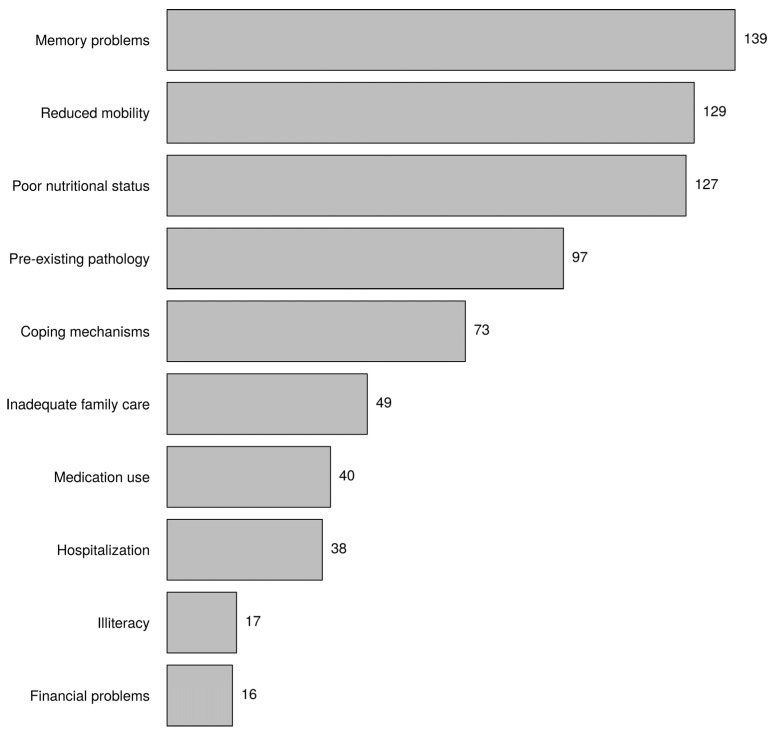
In your opinion, what are possible causes of frailty? Choose three answers (the biggest causes).

**Figure 5 healthcare-10-01632-f005:**
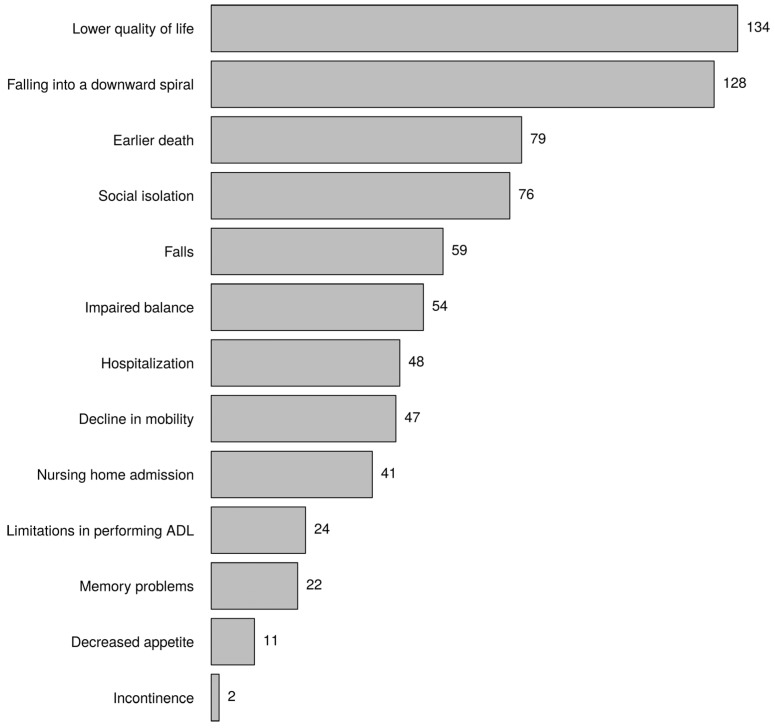
In your opinion, what are possible consequences of frailty? Choose three answers (the greatest consequences).

**Figure 6 healthcare-10-01632-f006:**
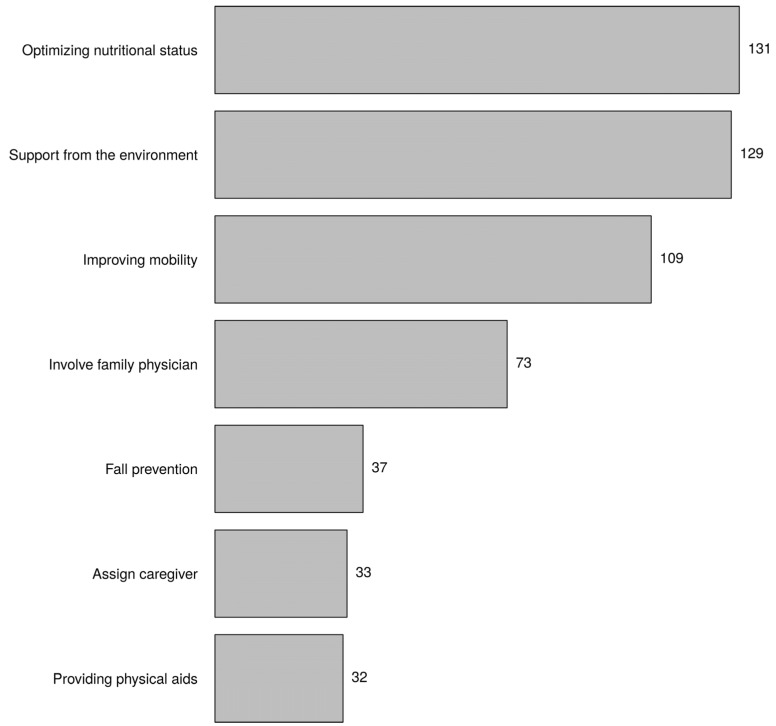
The process of frailty can be effectively reversed. What factors do you think have an influence on this process? Choose three answers (the factors that have the most influence).

**Figure 7 healthcare-10-01632-f007:**
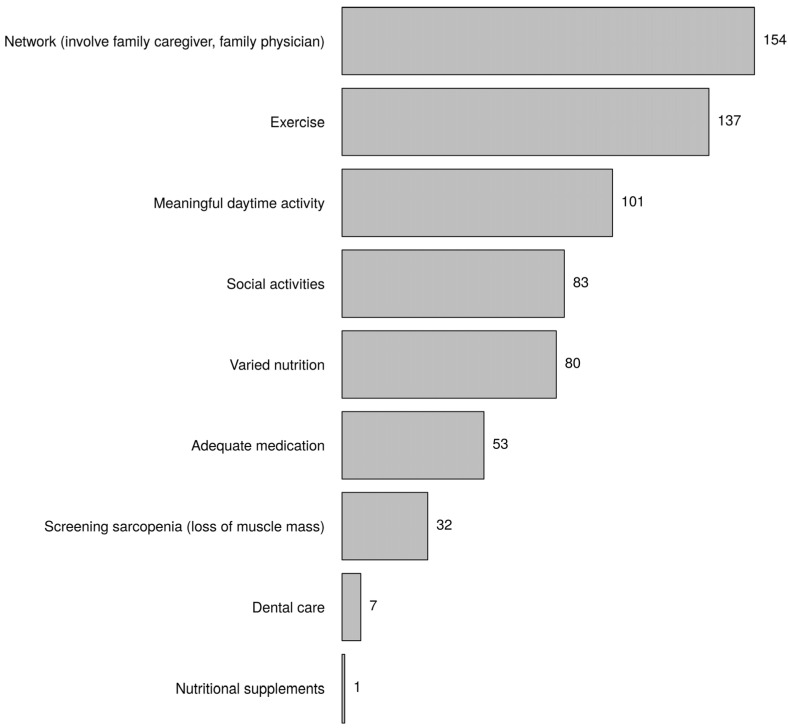
What preventive measures are you thinking of? Choose three answers (most appropriate preventive measures).

**Table 1 healthcare-10-01632-t001:** Characteristics of the respondents.

	Total	
	*N*	**%**
Characteristic		
**Gender**		
Male	14	5.6
Female	237	94.4
**Age category in years**		
<44	77	30.7
44 – <55	81	32.3
≥55	91	37.1
**Working experience in years**		
<20	70	27.9
20 – <33	90	35.9
≥33	91	36.2
**Working hours per week**		
<28	76	30.4
28 – <32	48	19.2
≥32	126	50.4
**Profession**		
Nurse	160	63.8
Nurse specialist	45	17.9
Other	46	18.3
Institution		
Hospital	105	41.8
Nursing home	65	25.9
Residential care centre	16	6.4
Primary healthcare	44	17.5
Psychiatric institution	3	1.2
Geriatric rehabilitation centre	9	3.6
Other	9	3.6
**Employed in years**		
<5	73	30.3
5 – <15	85	35.3
≥15	83	34.4
**Education**		
Post-secondary vocational education	49	19.7
Higher vocational education	123	49.4
Post-higher vocational education	6	2.4
Higher vocational education master	49	19.7
University	17	6.8
Other	5	2

## Data Availability

The data presented in this study are available on request from the corresponding author.

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
