# Peer review of "Nurses’ Opinions on Frailty"

_healthcare, 2022, doi:10.3390/healthcare10091632_

Round 1

Reviewer 1 Report

"Nurses’ Opinions on Frailty" is an interesting paper that shed light about the opinion of Nurses about frailty, which is a key concept for public health in the next years. The paper is informative and detailed, pleasant to the readers, and based on a sufficient amount of data.

In my opinion it highlights also the lack of nurses' propension to put at the centre of their action the social domains, together with other causes of frailty, even if many of them recognize this is a determinant of frailty. The interaction with social professional plays a crucial role in managing/preventing frailty at community level.  However, this lack of propension is not deeply discussed by the authors, that could improve this aspect of the paper.

Author Response

Dear Reviewer,

Thank you very much for your positive response to our manuscript.

The interaction with social professionals indeed plays an crucial role in preventing and managing frailty at community level. Based on your comment we added the following sentences to the discussion on page 17:

“In particular, if there are deficits in the social functioning of older people then collaboration with social workers plays an crucial role in managing frailty at community level. Factors that influence this collaboration are the perception of the other professional, closeness of the other professional, the perception of interdependence, and frailty of the older person [47].

47. Kim, M.-J. Factors affecting the collaboration between nurses in community health department and social workers in welfare services department.Health Policy and Management, 2008;18(4), 125-147.

Reviewer 2 Report

The purpose of this paper was to provide quantitative survey-based data regarding nurses' opinions on frailty. The authors argue that in order to provide early intervention for frailty, they must first understand the causes of frailty. However, this paper does not provide any educational interventions, but purely summarizes current conceptions of frailty. This paper would be greatly strengthened by providing an intervention or demonstrating how differences in opinions on frailty lead to differences in outcomes. As it stands, this paper does not appear to provide significant novelty to the current body of literature. 

Section 3.2 is confusing to read and should be separated into two paragraphs based on findings displayed in each figure. Figure 2 also appears to be missing. 

Section 3.3 indicates that all p-values were >0.01; was this meant to read <0.01? 

Statistical analysis could have been strengthened by performing multivariate analysis to determine if relationships between different variables impacted responses on frailty. 

Author Response

Dear Reviewer,

Thank you for your response to our manuscript.

The aim of study was to to explore the opinions of nurses working with older people on the concept of frailty, regardless of the care setting. The opinions were related to seven topics; prevention of frailty is only one of these topics. In addition, the associations between background characteristics of nurses and their opinions about frailty were examined. So providing an intervention or demonstrating how differences in opinions on frailty lead to differences in outcomes was not the aim of our study.

Only a few studies examined opinions of nurses on frailty [17, 18, 19]*. Our study is distinguished from these studies by its quantitative design. In addition, we examined associations between background characteristics of nurses and their opinions on frailty.

We agree with you that section 3.2 was difficult to read. Therefore, we have taken over your suggestion to make it two paragraphs.

Figure 2 is included in the manuscript.

With regard to section 3.3 there was no association between the characteristics of the respondents and the opinions on the frailty domains, so all p-values were >0.01.

Thank you for your suggestion to perform multivariate analysis. We agree with you. We therefore conducted some (backward) logistic regression analyses to see if they would lead to different insights. This was not the case. For that reason, we reported the results of the univariate analyses.

* References

17. Britton H. What are community nurses’ experiences of assessing frailty and assisting in planning subsequent interventions? British Journal of Community Nursing. 2017;22(9):440-5.

18. Papadopoulou C, Barrie J, Andrew M, Martin J, Birt A, Raymond Duffy FJ, et al. Perceptions, practices and educational needs of community nurses to manage frailty. British Journal of Community Nursing. 2021;26(3):136-42.

19. Warnier RMJ, van Rossum E, Du Moulin M, van Lottum M, Schols J, Kempen G. The opinions and experiences of nurses on frailty screening among older hospitalized patients. An exploratory study. BMC Geriatrics. 2021;21(1):624.

Reviewer 3 Report

This is a work of significant interest on a current topic.

It is well written, although the text needs to be revised since some paragraphs are not perfectly understandable and the use of some words could be improved. For example, dependency is sometime used instead of disability.

It is adequately justified, although, in the introduction, issues related to the different approaches to frailty that exist and the differences between them should be better justified.

Another problem is that I have not been able to see figure 2 of the job. I do not know if it is due to a technical problem or if it does not appear in the manuscript.

The main problem, in my opinion, is the questionnaire. The authors include in the discussion that it is not a validated questionnaire and assume this as a limitation. Of course. However, I think that some terms included in the questionnaire are not fully understood by themselves and should have been defined in material and methods and even included in questionnaire that was provided to the interviewees. If this was not done, it is possible that there may have been differences due to the lack of uniformity in the understanding of some terms. This aspect should be reviewed.

Author Response

Dear Reviewer,

Thank you very much for your positive response to our manuscript.

The text has been checked by a native speaker. So we assume that the English is good.

We have used the word “dependency” only once (on Page 14). In that sentence, we distinguish between disability and dependency.

Based on your comment on issues related to different approaches to frailty, we have added some information (see Page 2).

We have checked the presence of Figure 2 in our manuscript.

In response to your comment on the questionnaire we would like to clarify that the questionnaire was developed using focus groups and individual interviews. In addition, face validity of the questionnaire was examined among Dutch nurses. Also, all respondents had medical training and all worked in the field of frail older people. In our opinion, no terms were used in the survey that were not recognizable to the respondents.

Round 2

Reviewer 2 Report

Minor changes were made to the manuscript to improve readability, which is appreciated. Again, I believe this manuscript would provide much greater novelty if it related conceptions of frailty to patient outcomes, or proposed an educational intervention to solve this issue. 

Author Response

Dear Reviewer,

Thank you for your response to our revised manuscript.

The text has been checked by a native speaker. So we assume that the English is good.

In our opinion, the design of our study fits well with the aim of the study. The aim of our study was to explore the opinions of nurses working with older people on the concept of frailty. To achieve this aim, we administered a questionnaire to nurses. The novelty of our study concerns that there were no quantitative data available so far on the opinions of nurses about topics related to frailty (keywords of frailty, frailty domains, causes of frailty, consequences of frailty, reversing frailty, the prevention of frailty, and addressing frailty). Achieving the aim you have in mind requires a different design.

However, we agree with you that it is important to examine the effects of nurses’ opinions about frailty on patient outcomes. In addition, based on the results of our study the development and implementation of an educational intervention can be recommended. Based on your comment we added the following paragraph to the discussion:

“Our study shows the opinions of nurses on frailty. It has become clear that nurses view frailty differently. What this study did not examine was the effect of this on the quality of care and outcomes in older people (e.g. an increase in healthcare utilization, disability, lower quality of life, premature death). It would be interesting and important to conduct a study focused on the associations between nurses' opinions about frailty and these outcomes. In addition, we recommend to develop an educational intervention based on the results of this study. This will potentially lead to a higher level of knowledge about frailty among nurses. Follow-up research can determine the effects of the developed educational intervention on adverse outcomes in older people, for example by performing a randomized controlled trial”.